# Maternal Employment Status and Minimum Meal Frequency in Children 6-23 Months in Tanzania

**DOI:** 10.3390/ijerph16071137

**Published:** 2019-03-29

**Authors:** Lauren C. Manzione, Heidi Kriser, Emily G. Gamboa, Curtis M. Hanson, Generose Mulokozi, Osiah Mwaipape, Taylor H. Hoj, Mary Linehan, Scott Torres, P. Cougar Hall, Josh H. West, Benjamin T. Crookston

**Affiliations:** 1Department of Public Health, Brigham Young University, LSB, Provo, UT 84602, USA; laurenm.byu@gmail.com (L.C.M.); kriserheidi@gmail.com (H.K.); gamboa.emily101@gmail.com (E.G.G.); curtisha.byu@gmail.com (C.M.H.); taylorhoj@gmail.com (T.H.H.); coughall@gmail.com (P.C.H.); West.Josh@gmail.com (J.H.W.); 2IMA World Health, Nyalali Curve, PO Box 9260, Plot 1657, Dar es Salaam, Tanzania; generosemulokozi@imaworldhealth.org (G.M.); osiahmwaipape@imaworldhealth.org (O.M.); marylinehan@imaworldhealth.org (M.L.); scotttorres@imaworldhealth.org (S.T.)

**Keywords:** minimum meal frequency, childhood nutrition, Tanzania, maternal employment, childcare practices

## Abstract

As women in developing world settings gain access to formal work sectors, it is important to understand how such changes might influence child nutrition. The purpose of this paper is to examine the relationship between maternal employment status and minimum meal frequency (MMF) among children in Tanzania. Interviews were conducted with 5000 mothers of children ages 0–23 months. The questionnaire used in these interviews was developed by adopting questions from Tanzania’s latest Demographic and Health Survey (2015–2016) where possible and creating additional questions needed for programmatic baseline measurements. MMF was used as proxy for child nutrition. Logistic regression analyses were used to identify associations between employment status and parenting practices of Tanzanian mothers and MMF of their children. After adjusting for confounders, informal maternal employment [OR = 0.58], lack of financial autonomy [OR = 0.57] and bringing the child with them when working away from home [OR = 0.59] were negatively associated with meeting MMF. Payment in cash [OR = 1.89], carrying food for the child [OR = 1.34] and leaving food at home for the child [OR = 2.52] were positively associated with meeting MMF. Informal maternal employment was found to be negatively associated with meeting MMF among Tanzanian children. However, behaviors such as bringing or leaving prepared food, fiscal autonomy and payment in cash showed significant positive associations. These findings could help direct future programs to reduce child stunting.

## 1. Introduction

Undernutrition is a contributing cause for more than one-third of the approximately six million child deaths that occur annually across the world [1,2]. Stunting is the most prevalent type of undernutrition and it continues to be a major public health problem [3]. Global statistics from the World Bank indicate that 22.2% of children worldwide aged less than 5 years are stunted [4]. Low- and middle-income countries are disproportionately affected, with more than 150 million stunted children under 5 years of age [5]. 

The prevalence of undernutrition is highest in Sub-Saharan Africa, where approximately 42% of children are considered stunted (height-for-age z-score two standard deviations below median) [6]. Trend analyses project an upward trend until the year 2020 within East Africa, the African continent, and worldwide [3]. In Tanzania the stunting prevalence rate is approximately 34%, making it the third highest in Sub-Saharan Africa, as well as one of the top ten worst affected countries in the world [2,7,8].

Due to the severity of the effects of undernutrition, particularly for children younger than five, stunting is a major health concern in Tanzania [2,6]. Stunting not only restricts growth, but it also is a contributing cause for morbidity and mortality [2]. Additionally, it can result in lower physical capacity, reduced productivity, decreased cognitive function, and impeded school performance [3,7,9,10,11,12]. These conditions, then, have the likelihood to perpetuate the cycle of poverty, especially in countries like Tanzania where high percentages of children are affected [3,9].

Risk factors for stunting are multifactorial and can vary depending on location [3], including low socioeconomic status, low dietary diversity, low minimum meal frequency (MMF)—defined as the proportion of the population who eat the recommended minimum number of meals in a day—infections, poor sanitation, and inadequate health care services [2,13,14,15,16,17,18]. The care-giving behaviors of the primary caregiver, which in Tanzania is usually the mother, is also a contributing factor. For example, women’s autonomy as it relates to social status and decision-making power within the home, nutritional status during pregnancy and lactation, education level, age, breastfeeding, and young child feeding behaviors, and employment status all impact on rates of undernutrition among their children [2,3,5,7,13,16,17,18].

As a primary risk factor for stunting, MMF among children in Tanzania has been the focus of previous research. A study exploring inappropriate complementary feeding practices among children 6–23 months in Tanzania found that only 38.6% of children met MMF guidelines [19]. In this study, meeting MMF guidelines was positively associated with greater asset ownership, accessing post-natal care, and gender, with male children reaching MMF guidelines significantly more than female children [19]. While in this study MMF was not found to be associated with maternal education, children whose mothers had paid work were significantly more likely to meet MMF recommendations than those mothers who were not working [19]. In a similar study including mothers of children 6–24 months in Tanzania, however, the prevalence of stunting was higher in children whose mothers worked full-time [20]. The nearly 45% of full-time working mothers worked just over 10 hours per day. The majority of these women left their child at home with relatives or hired other caregivers (84%), while 16% of study mothers worked and looked after their child simultaneously. This study found low rates of exclusive breastfeeding (9%) and frequent early introduction of complementary foods, this study concluded that maternal employment presents significant challenges to optimal infant and child feeding practices in Tanzania [20]. Additional research is needed to understand the complex relationship between maternal employment and child feeding practices associated with stunting prevention, particularly MMF. The purpose of this study is to examine the employment status and related behaviors of mothers and their associations with MMF among children ages 6–23 months in Tanzania.

## 2. Materials and Methods

### 2.1. Design

This study uses data derived from a cross-sectional baseline survey conducted in January and February of 2016 for IMA World Health’s “Addressing Stunting in Tanzania Early” (ASTUTE) project. The project is implemented in five lake zone regions of Tanzania and seeks to reduce stunting among children aged less than 5 years. The data was gathered to inform program development and establish a baseline from which to measure program impact.

### 2.2. Sample

This study sample is composed of 5000 female primary caregivers of children ages 0 to 23 months. Respondents were recruited from five regions including Geita, Kagera, Kigoma, Mwanza, and Shinyanga. Probability proportional to size sampling was used down to the district level, with the most recent (2012) Tanzania census as a sampling frame. Specific villages or streets were then randomly selected. If a mother was not home, three attempts were made to contact households before replacement households were used.

### 2.3. Procedure

One-hour face-to-face interviews were conducted in Kiswahili. The survey instrument was field-tested, revised, and finalized prior to administration by Ipsos Tanzania. The questionnaire was scripted onto a mobile data collection platform and uploaded to Android mobile devices used for data collection. Informed consent was acquired from all study participants, and the National Institute for Medical Research in Tanzania and relevant local government authorities authorized the research (NIMR/HQ/R.8a/Vol.IX/2344). Three research teams, trained by Ipsos Tanzania, administered the finalized questionnaire. Refusals to participate totaled 150 among the five designated regions. Upon completion of data collection, Ipsos Tanzania compiled survey results for cleaning and analysis. 

### 2.4. Measurement and Analysis

Demographic information included respondent’s age, education level, writing and reading ability, religion, marital status, number of children, child’s age, and an asset index representing wealth. Maternal occupational status, behavioral practices, and situational characteristics were also recorded. The asset indicator was created by summing the number of assets respondents had indicated they owned out of 21 possible assets. These assets included the following: adult bicycle, motorcycle, car or truck, animal drawn cart, boat with motor, boat without motor, radio, television, mobile phone, refrigerator, table, chairs, bed, air conditioner, computer, electric iron, fan, power tiller, connection to the national electricity grid, active mobile banking account (e.g., M-PESA), and owns more than one acre of agricultural land.

Occupational status included 21 potential categories and was then recoded into three categories including: (1) not employed, (2) informally employed, and (3) formally employed. Not employed included mothers who were unemployed/not looking for work, a housewife, or an unpaid family helper. Informally employed included occupations related to farming, self-employed occupations, and paid family helpers. Formally employed included occupations related to governmental jobs, public/private companies, or non-governmental organization/religious occupations. Students, retired individuals, those searching for a job, and those incapable of working were excluded because they occurred rarely and were distinctly different from the three main categories.

Questions highlighting behavioral and situational behaviors of respondents were considered in relation to their potential influence on childhood feeding practices. Employment-related feeding practices were identified using the following questions: “When you work outside the home or are away from home, do you take your child with you?”, “Are you paid in cash or (in-)kind for this work or are you paid at all?”, “Who usually decides how the money you earn will be used?”, “If you take your baby with you, do you carry food for him?”, “If you leave the child at home, do you prepare food in advance to be given to your child while you are away?”, and “Who watches the child while you are away?”.

The World Health Organization defines a meal as receiving solid, semi-solid, or soft foods (also including milk for children who are not breastfed) for children 6–23 months old. This definition stipulates the minimum amount of meals, which is as follows: two meals per day for infants 6–8 months, three meals per day for breastfed children 9–23 months, and four meals per day for non-breastfed children 6–23 months [18]. Children who met this definition of MMF were coded as ‘yes’, all other responses were coded as ‘no.’ Responses were collected from those with children ages 0–23 months, but those younger than 6 months were excluded from the study.

All statistics were calculated using SAS version 9.4 by researchers with significant statistical training. Descriptive statistics were calculated for demographic variables. Logistic regression analysis was used to determine associations between primary caregiver occupation status and MMF among their children, while controlling for basic demographic factors. All regression models included maternal marital status, maternal age, maternal education level, and household asset indicator. Logistic regression analysis was also used to determine associations between behavioral variables in relation to mothers and MMF among their children.

## 3. Results

### 3.1. Characteristics of Respondents

The average age of respondents was 27.5 years old (Table 1). The majority of mothers completed only primary school (56.16%) with almost one-third not completing primary education (30.24%). The average number of assets owned was 4.8, and the majority of respondents had a husband or partner (85.3%).

Exactly 82% of respondents’ children met minimum meal frequency (Table 2). The majority of mothers were informally employed (80.3%) and paid in cash (62.7%). Most mothers reported bringing their children to work with them at least some of the time, with 40.4% bringing them all of the time. The majority of respondents did not bring food for their child when they were away from home (51.0%) or leave food behind when they left their child at home (58.2%).

### 3.2. Predictors of Minimum Meal Frequency

After adjusting for potential confounders, maternal employment (informal vs. not employed) [OR = 0.58; CI = 0.14–0.80], financial decision-making (husband/partner vs. respondent) [OR = 0.57; CI = 0.35–0.93], and bringing the child with them (most of the time vs. never [OR = 0.59; 0.41–0.85]; all of the time vs. never [OR = 0.57; CI= 0.40–0.82]) were all negatively associated with achieving MMF. Payment method (paid in cash vs. not paid) [OR = 1.89; CI = 1.25–2.86], carrying food for the child (yes vs. no) [OR = 1.34; CI = 1.04–1.74], and leaving food for the child (yes vs. no) [OR = 2.52; CI = 2.04–3.11] were positively associated with meeting MMF (Table 3).

## 4. Discussion

The purpose of this study was to examine employment status and related behaviors of mothers and their associations with MMF among children ages 6–23 months in Tanzania. Results indicate that children of respondents that were informally employed were less likely to meet MMF guidelines. These findings are consistent with other studies, which also showed association between maternal employment and suboptimal child nutrition [2,18,21].

### 4.1. Maternal Employment

There are multiple potential explanations for the relationship between maternal employment and child nutrition. A 2017 study of children ages 6–59 months in Southern Ethiopia found no statistical association between maternal employment and nutrition status, however this population has low rates of stunting overall [5]. In contrast, maternal employment was negatively associated with the nutrition status of children in Bangladesh. It is unclear if these factors might also be at play in the current study, but there are some important socioeconomic similarities between Tanzania and Bangladesh. A study exploring factors associated with child malnutrition in poor areas of China found that child nutrition was dependent on the type of a mother’s employment [22]. Children of mothers considered professional personnel or assigned positions in the military were significantly less likely to be malnourished as compared to children of mothers who were farmers, rural handicraft laborers, or who individually owned their own business [22]. Findings from a study in Nepal were similar, indicating that children of mothers engaged in informal work (e.g., manual laborers, street venders) were at significantly greater risk of being malnourished than children of mothers engaged in formal work [23]. Farming as an occupation in the current study was coded as informal employment, which was associated with inadequate feeding practices. The benefits of more formal employment, including better income, improved working conditions, and increased access to resources may help to explain these findings.

Results indicate that those caring for children and working simultaneously were less likely to have a child meet MMF guidelines. There may be many explanations for this finding. A study including female construction workers in India identified fear of wage loss due to decreased productivity as a primary obstacle in attending to a child’s needs when accompanying mother to work [24]. In other words, even when a mother or other caregiver is permitted to bring her child to work, she may feel pressure to neglect a child’s nutritional needs in order to maintain employment and provide financially for her family. In the current study, however, bringing food from home for children to eat at work was positively associated with meeting MMF guidelines. While the age of the alternate care provider at home with a child in the current study was not significant, results did indicate that a caregiver’s leaving food for their children at home before going to work is a practice positively associated with meeting MMF guidelines. This practice may provide a workable and effective solution for caregivers who both need to work outside the home and yet cannot take their child and food for the day with them to work.

### 4.2. Maternal Decision Making Power

Results indicate that women who have less decision-making power in their household were less likely to meet MMF guidelines. While gainful employment has been positively associated with better nutrition, positive nutritional outcomes are also often related to higher levels of women’s status and decision-making power [5]. The current study identified that among households where the husband or male partner was primarily responsible for how money was spent, the children were less likely to meet MMF guidelines compared to instances where the primary caregiver either had total control or shared decision making. This finding may support the notion that employment’s impact on feeding practices is, at least in part, related to increased power associated with employment. Gender inequality and low female empowerment likely impact childhood nutrition in a number of ways. For example, findings from focus-group discussions with women in rural Gambia indicate that while women are often the sole caretakers of their children with little or no assistance from their husbands, household decisions are either strongly influenced or made entirely by men. With little autonomy or agency, women are expected to take care of household chores and farm work, minimizing the time they have for the health and nutritional needs of themselves or their children [25].

### 4.3. Maternal Employment Compensation

The current study also considered how mothers are paid for their labor. Children of mothers paid in cash rather than not paid were more likely to meet MMF. Almost one-third of the women in the current study were employed in some way but not paid in cash or kind, for example, women farming their own land for food to sell, for instance, were considered employed without being paid. These findings support the conclusion that more formal maternal work conditions are generally associated with improved health and nutritional outcomes for children.

### 4.4. Employment and Child Care

In the current study, a majority of participants who leave children at home when going to work do not leave food for their children. These mothers may lack the self-efficacy to prepare and leave food that will not quickly spoil for their children or lack the time and resources to do so. Their perceptions of what is socially acceptable may also make it commonplace to leave children at home with other children less than 12 years of age, which was true of almost one-third of the sample. This could be in large part because parents may have no other options for child care. Furthermore, social perceptions regarding women’s financial autonomy may also create a barrier to self-efficacy in managing finances.

### 4.5. Future Research

Future studies should utilize stunting outcomes directly as calculated using height-for-age z-scores, providing a more direct comparison of factors influencing rates of stunting among Tanzanian children. Future research should also address what social norms, barriers, or feelings of self-efficacy might impact a mother’s ability to improve her behavioral lifestyle choices. Future programs aimed at reducing stunting through maternal employment should recognize the value of motivating mothers to bring food to work or to prepare food ahead to leave food for children, and to try to receive payment in cash. Such behavioral changes can significantly improve MMF among children ages 6-23 months, encouraging a reduction in stunting rates in Tanzania.

### 4.6. Limitations

Several limitations may have impacted findings from the current study. One such limitation is related to the categorization of occupation status. In defining occupation status among participants, the response options were not necessarily mutually exclusive, though only one response could be given. For example, a woman could consider herself a housewife and a crop farmer if she is cultivating her own land. The employment variable used for analysis could thus be improved to better represent the occupational and employment status of survey participants who gave one answer though they could have two occupations. Another limitation to consider is that data was gathered in villages from five Tanzanian regions and consequently may not be representative of the general population in Tanzania. Furthermore, the ability to achieve MMF may change seasonally throughout Tanzania due to the availability of food, where food is more abundant after the rainy season, and less available during planting season. This survey was conducted in January and February, and may therefore be representative of that season specifically, thus not accurately reflecting the seasonal variation of food throughout the year. Other considerations include a limited number of formally employed participants to allow for a comparison of informal versus formal employment status in relation to observed outcomes. However, the demographic situation of study regions may not currently lend itself to any significant measure due to a very small portion of the population being formally employed. Additionally, one potential bias in this study is a social desirability bias. Social desirability bias drives an individual to answer in a way that makes them look more favorable to the interviewer, others in the room, or to society in general. Caregivers may have felt that they should respond in a way that was pleasing to the interviewer, which could have biased the results. Also, there were some instances in which husbands/male partners desired to sit in and listen during the interview, which also could have influenced respondents’ readiness to participate and respond freely. This situation was relatively rare, as efforts were made to interview the caregiver without her husband/male partner present, resulting in only 4.62% (231/5000) of interviews being conducted with a husband/male partner present. Lastly, some individuals refused to participate in the interview due to being busy with cultivation activities, not fully understanding the purpose of the interview, or other unknown reasons. This was, however, relatively infrequent, with a total of 150 refusals occurring over the course of the data collection process.

## 5. Conclusions

Achieving MMF is associated with women’s ability to get paid, rely on adequate child care support, and attain greater gender equity. Potential interventions for improving MMF in Tanzania could include supporting mothers who bring their child to work by helping mothers develop plans and strategies to bring food for the child with them. In cases where a mother works and does not have financial autonomy over the money she earns, focusing on household strategies that address gender equity may be effective. Interventions might include developing awareness and strategies to encourage parents to leave their child with someone who is at least 12 years of age. Additionally, if a woman must leave her child at home, interventions might focus on the preparation of non-perishable food that can be consumed by the child while she is away.

## Figures and Tables

**Table 1 ijerph-16-01137-t001:** Participant Demographics.

Demographics	N (%)/Mean (SD)
Mean age of respondent (SD)	27.5 (7.3)
Mean Asset Index * (SD)	4.8 (2.2)
Education status	
None/Primary incomplete	1546 (30.9%)
Complete primary	2808 (56.2%)
Some secondary or more	646 (12.9%)
Relationship status	
Single	735 (14.7%)
Has husband/partner	4265 (85.3%)

* Wealth Index includes a sum of all the commodities owned by the participant. Commodities include: adult bicycle, motorcycle, car or truck, animal-drawn cart, boat with motor, radio, television, mobile phone, refrigerator, table, chairs, bed, air conditioner, computer, electric iron, fan, power tiller, connection to national electricity grid, active mobile banking account, owns more than one acre of agricultural land.

**Table 2 ijerph-16-01137-t002:** Key Respondent Characteristics.

Indicator	Frequency	%
Met Minimum Meal Frequency		
Yes	2412	82.0
No	530	18.0
Main occupation		
Not employed	797	16.0
Informally employed	3996	80.3
Formally employed	182	3.7
Employment payment		
Not paid	387	31.0
Paid in kind	16	1.3
Paid in cash	781	62.7
Paid in cash and in-kind	62	5.0
Took child to employment		
Never	801	16.0
Some of the time/Rarely	1168	23.4
Most of the time	1009	20.2
All of the time	2022	40.4
Took food for child when away		
No	2145	51.0
Yes	1049	25.0
Exclusively breastfed	1005	23.9
Age of child caregiver when away		
<12 years old	467	23.7
≥12 years old	1502	76.3
Prepare food in advance for child when away		
No	2893	58.2
Yes	2081	41.8
Who makes decisions about money respondent earns		
Respondent	576	48.2
Husband/partner	432	36.2
Respondent and husband/partner jointly	187	15.7

**Table 3 ijerph-16-01137-t003:** Regression Analysis for Minimum Meal Frequency and Employment Status.

Association between Minimum Meal Frequency and Main Occupation		
*Main occupation*	*Unadjusted Odds Ratio (CI)*	*Adjusted* Odds Ratio (CI)
Not employed	––	––
Informally employed	0.55 (0.41–0.74) ***	0.58 (0.41–0.80) *
Formally employed	1.27 (0.64–2.52)	1.18 (0.55–2.52)
Association between Minimum Meal Frequency and Employment Payment Type		
*How paid*	*Unadjusted Odds Ratio (CI)*	*Adjusted* Odds Ratio (CI)
Not paid	––	––
Paid in kind	2.40 (0.28–20.33)	2.26 (0.26–19.79)
Paid in cash	2.00 (1.37–2.90) **	1.89 (1.25–2.86) *
Paid in cash and in kind	2.47 (0.99–6.13)	2.21 (0.87–5.62)
Association between Minimum Meal Frequency and Money Decision Maker		
*Money decision maker*	*Unadjusted Odds Ratio (CI)*	*Adjusted* Odds Ratio (CI)
Respondent	––	––
Husband/partner	0.55 (0.37–0.82) *	0.57 (0.35–0.93) *
Jointly	0.80 (0.46–1.39)	0.73 (0.40–1.37)
Association between Minimum Meal Frequency and Child taken to Employment		
*Took child to employment*	*Unadjusted Odds Ratio (CI)*	*Adjusted* Odds Ratio (CI)
Never	––	––
Sometimes/Rarely	0.77 (0.54–1.10)	0.80 (0.56–1.17)
Most of the time	0.51 (0.36–0.72) **	0.59 (0.41–0.85) *
All of the time	0.55 (0.39–0.76) **	0.57 (0.40–0.82) *
Association between Minimum Meal Frequency and Carry Food for Child		
*Carry food for child when away*	*Unadjusted Odds Ratio (CI)*	*Adjusted* Odds Ratio (CI)
No	––	––
Yes	1.40 (1.10–1.78) *	1.34 (1.03–1.73) *
Association between Minimum Meal Frequency and Age of Child Caregiver		
*Age of child caregiver when away*	*Unadjusted Odds Ratio (CI)*	*Adjusted* Odds Ratio (CI)
<12 years old	––	––
≥12 years old	1.45 (1.03–2.06) *	1.33 (0.91–1.93)
Association between Minimum Meal Frequency and Food Preparation		
*Prepare food for child when away*	*Unadjusted Odds Ratio (CI)*	*Adjusted* Odds Ratio (CI)
No	––	––
Yes	2.70 (2.22–3.29) ***	2.52 (2.04–3.11) ***

* *p* < 0.05; ** *p* < 0.001; *** *p* < 0.0001. Note: All adjusted models include maternal marital status, maternal age, maternal education level, and household asset indicator.

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
