# Peer review of "Maternal Employment Status and Minimum Meal Frequency in Children 6-23 Months in Tanzania"

_ijerph, 2019, doi:10.3390/ijerph16071137_

Round 1

Reviewer 1 Report

Abstract: well written. However, authors didn’t mention the tool used to collect the data, for example; validated questionnaire. 

Introduction: (page 1, line 31) trends analyses project... this sentence needs more elaboration.

(Page 2, line2) please define MMF. 

Methods: 

(Page 2, sample) which age group? It was mentioned in the aim of the study that the age group was 6-23 months old? Or is it 0-23 months? 

(Page 3, line 25) please define abbreviations NGO....etc

(Page 3, line 41) majority of children (82%)... this should be in results section.

(Page 3, line 42) statistics.... I suggest that the authors add if a statician was involved in the analyses.

In addition, I suggest to whether there were any inclusion/exclusion criteria.

Discussion: (Page 6, line14): similarities between Tanzania and Bangladesh.... I suggest that the authors compare and contrast between Tanzania and other countries from the same geographical region (African countries).

Limitations: I suggest that the authors add the disadvantages or possible bias of using this questionnaire.

Conclusion: well written. 

Author Response

Reviewer

Concern

Response

Edit/Location

1

Abstract: well written. However, authors didn’t mention the tool used to collect the data, for example; validated questionnaire. 

The following sentence was added to the abstract to add this information:

“The questionnaire used in these interviews was developed by adopting questions from Tanzania’s latest Demographic and Health Survey (2015-2016) where possible and creating additional questions needed for programmatic baseline measurements.”

Abstract, sentence 4

1

Introduction: (page 1, line 31) trends analyses project... this sentence needs more elaboration.

“Trend analyses project an upward trend until the year 2020 within East Africa, the African continent, and worldwide”

Page 1

1

(Page 2, line2) please define MMF. 

“defined as the proportion of the population who eat the recommended minimum number of meals in a day” a more in-depth definition is given on page 3, lines 38-42

Page 2, lines 2-5

1

(Page 2, sample) which age group? It was mentioned in the aim of the study that the age group was 6-23 months old? Or is it 0-23 months? 

The questionnaire was given to mothers of children ages 0-23 months, but our analyses only included mothers of children 6-23 months. We elaborated more in the measurement and analysis section to avoid confusion.

Page 2 (sample)

Page 4, line 4 (Measurement and analysis)

1

(Page 3, line 25) please define abbreviations NGO....etc

Non-Governmental Organization

Page 3, Line 27

1

(Page 3, line 41) majority of children (82%)... this should be in results section.

Removed from methods and included in results.

Removed from page 3, line 44 and included on page 4, line 20.

1

(Page 3, line 42) statistics.... I suggest that the authors add if a statistician was involved in the analyses.

All statistics were calculated using SAS version 9.4 by researchers with significant statistical training.

Page 4, line 5

1

In addition, I suggest to whether there were any inclusion/exclusion criteria.

“Students, retired individuals, those searching for a job, and those incapable of working were excluded because they occurred rarely and were distinctly different from the three main categories.” 

Responses were collected from those with children ages 0-23 months, but those younger than 6 months were excluded from the study.”

Page 3, lines 28-30

Page 4, starting on line 4

1

Discussion: (Page 6, line 14): similarities between Tanzania and Bangladesh.... I suggest that the authors compare and contrast between Tanzania and other countries from the same geographical region (African countries).

“A 2017 study of children ages 6-59 months in Southern Ethiopia found no statistical association between maternal employment and nutrition status, however this population has low rates of stunting overall [29]. 

Starting page 6, line 10

1

Limitations: I suggest that the authors add the disadvantages or possible bias of using this questionnaire.

Several sentences were added to elaborate on the potential biases/disadvantages of the questionnaire/data collection.

Page 9, end of limitations section

Reviewer 2 Report

Contribution of interest and attempts to address an important question. Generally well done with several strengths.

Some specifics.

Was household asset indicator associated with  the MMF? Further, contributory value of this work will be enhanced if how such HH level macro variable interacted with say mother level predictors of MMF.

Was there any heterogeneity of results by child age? Such mothers of younger child vs older or child age gradations in observed associations and other interpretations?  Mothers of younger children may be precluded from certain occupations for example, which could have an impact on MMF.

Some basic IYCF feeding characteristics will help readers relate to this work. E.g. exclusive breasting, time to breastfeeding after birth etc.

Table 3. Would be good if potential confound variables are listed as footnote to guide the reader

Ref # 27 incorrect, i think, citation numbers might have moved around - please check all.

Author Response

Reviewer

Concern

Response

Edit/Location

2

Was household asset indicator associated with the MMF? Further, contributory value of this work will be enhanced if how such HH level macro variable interacted with say mother level predictors of MMF. 

Those with more assets were significantly more likely to achieve MMF than those with fewer assets. We felt that since SES has been well established as an indicator of health status it would serve as fitting confounding variable but did not go into more detail about the results in the paper. Our future research will look at interactions between HH data and other variables more in-depth. Thank you for the suggestion.

2

Was there any heterogeneity of results by child age? Such mothers of younger child vs older or child age gradations in observed associations and other interpretations? Mothers of younger children may be precluded from certain occupations for example, which could have an impact on MMF.

Those who achieved MMF had higher odds of being age 12-17 months and 18-23 months than those 6-12 months. It is certainly possible that there could be interactions between child’s age and other factors. It is also assumed that those ages 6-12 are more likely to still be breastfeeding than older children.

2

Some basic IYCF feeding characteristics will help readers relate to this work. E.g. exclusive breasting, time to breastfeeding after birth etc. 

Table 2 provides some information on what percentage of the population of interest exclusively breastfed their child. We’ve highlighted this in Table 2 for easy reference. Regarding the indicator of time to breastfeeding after birth (timely initiation of breastfeeding), we chose not to include this and other related indicators because our main outcome variable is minimum meal frequency and is limited to children 6-23 months of age.

Table 2

2

Table 3. Would be good if potential confound variables are listed as footnote to guide the reader

****All adjusted models include maternal marital status, maternal age, maternal education level, and household asset indicator.

Page 6, table 3 footnote

2

Ref # 27 incorrect, I think, citation numbers might have moved around - please check all. 

References were mixed up during editing. All the references have been corrected.

References

Round 2

Reviewer 1 Report

Please include all your corrections and response in the main text.